# Optimizing Cocoa Productivity Through Soil Health and Microbiome Enhancement: Insights from Organic Amendments and a Locally Derived Biofertilizer

**DOI:** 10.3390/microorganisms13061408

**Published:** 2025-06-17

**Authors:** Jennifer E. Schmidt, Julia Flores, Luigy Barragan, Freddy Amores, Sat Darshan S. Khalsa

**Affiliations:** 1Plant Science Center, Mars Wrigley, Davis, CA 95616, USA; 2La Chola Farm, Mars Wrigley Science & Technology, Guayaquil 091003, Ecuador; 3Department of Plant Sciences, University of California Davis, Davis, CA 95616, USA

**Keywords:** biofertilizer, compost, organic matter, plant growth-promoting rhizobacteria (PGPR), soil amendment, soil health, *Theobroma cacao*, vermicompost

## Abstract

Despite growing interest in improving soil health on cocoa farms, applied research on the impacts of specific amendments on soil and plant outcomes is lacking. An integrated assessment of the impacts of two different organic amendments (compost and vermicompost) and a microbial biofertilizer on soil physical, chemical, and biological properties, as well as cocoa flowering, fruit set, and yield, was conducted in Guayaquil, Ecuador. Complementary culture-dependent and culture-independent methods were used to assess the impacts of amendments on microbial diversity, community composition, and specific taxa. Compost or vermicompost application affected soil chemical properties, including potassium, phosphorus, and sodium, and had small but significant effects on fungal beta diversity. Biofertilizer application slightly lowered soil pH and altered the total abundance of specific taxonomic groups including *Azotobacter* sp. and *Trichoderma* sp., with borderline significant effects on *Azospirillum* sp., *Lactobacillus* sp., *Pseudomonas* sp., calcium-solubilizing bacteria, and phosphorus-solubilizing bacteria. Amplicon sequencing (16S, ITS) identified 15 prokaryotic and 68 fungal taxa whose relative abundance was influenced by organic amendments or biofertilizer. Biofertilizer application increased cherelle formation by 19% and monthly harvestable pod counts by 11% despite no impact on flowering index or annual pod totals. This study highlights the tangible potential of microbiome optimization to simultaneously improve on-farm yield and achieve soil health goals on cocoa farms.

## 1. Introduction

The production of cocoa (*Theobroma cacao* L.) must evolve in order to meet increased demand while also prioritizing environmental sustainability. Maintaining or improving soil health over the long term is a key component of this evolution, and doing so will require optimizing the physical, chemical, and biological properties of soil.

Adding organic amendments such as compost, vermicompost, or manure to the soil is a widespread practice undertaken to provide nutrients and build soil organic matter (SOM). Reviews in annual crops have shown that these amendments release mineralized nutrients and can improve yield not only in the current growing season [1] but also in future seasons if they effectively increase SOM [2]. Vermicompost may provide an advantage over compost in tropical soils in this regard as the vermicompost process can increase the stability of organic matter (OM) in tropical climates [3]. Building SOM also helps improve the water-holding capacity and sequesters carbon, contributing to agroecosystem resilience and climate change mitigation [4,5]. While organic amendments alone may not improve yield in intensively fertilized annual systems in temperate climates [6], they may provide additional benefits in combination with inorganic fertilizer in tropical soils in specific conditions [2].

In research conducted on cocoa farms, the effects of OM on cocoa growth and yield have depended significantly on the qualities of those amendments and the farm context, ranging from no effect to impressively large increases in yield. A 3.5-year study assessing the potential of soil amendments and fertilization to rehabilitate a 22-year-old degraded cocoa plantation found that applying compost at a rate of 4.4–5.5 MT/ha/yr with an organic fertilizer increased pod yield relative to organic fertilizer alone, but adding compost to mineral fertilizer did not increase pod yields [7]. In a study of cocoa seedlings under greenhouse conditions, only cocoa pod compost enriched with poultry manure or inorganic P fertilizer increased seedling dry biomass, whereas compost enriched with N and unenriched compost had no effect [8]. In another greenhouse experiment, the application of high rates (80 MT/ha) of rice husk compost resulted in decreased cocoa seedling growth, perhaps because the high C:N ratio resulted in nutrient immobilization [9]. At the other end of the spectrum for effect size, dry bean yield were increased fivefold by application of 10 kg compost per tree per year as compared to mineral fertilizer or dolomite in a field trial on marginal soil in Sulawesi, Indonesia [10]. While very few studies have looked at the effects of OM on microbial communities in cocoa-growing soils, Doungous et al. found that application of cocoa pod husk compost increased biological activity, as well as populations of actinomycetes and fungi [11]. Affordability, availability, and return on investment of different forms of OM also play a role in the decision-making process for cocoa growers. There is a need for more research that specifically integrates those economic considerations into assessments of the impacts of different organic amendments on soil physical, chemical, and especially biological properties and cocoa productivity.

Inoculation with plant growth-promoting biofertilizers, also called efficient microorganisms (EM), is also increasingly popular as a strategy to increase crop yield and potentially soil health [12,13]. Biofertilizer development has changed over time and multiple types of biofertilizers are now common. Single-strain biofertilizers derived from a naturally occurring plant growth-promoting microorganism are common in commercial formulations, while growing attention is devoted to synthetic communities (SynComs) of multiple synergistic strains and microbiome engineering strategies such as gene editing to optimize target traits [14]. Locally derived EM blends, in which autochthonous microorganisms are isolated, propagated, and reapplied to augment beneficial taxa already present, also provide an alternative to introducing exogenous strains.

A number of inoculation studies of biofertilizers in cocoa were reviewed by Schmidt et al. [15]. Among fungi, the majority belonged to various genera of mycorrhizae [16,17], with one study testing *Trichoderma hamatum* [18]. Prokaryotic biofertilizers, whose origins varied by study, included isolates of *Bacillus* sp. [19,20,21], *Streptomyces cameroonensis* [22], *Enterobacter cloacae* [21], and a mixture of *Azospirillum brasilense, Chromobacterium violaceum,* and *Acinetobacter calcoaceticus* [23]. Other studies not included in that review have tested additional potential plant growth-promoting strains, finding benefits for strains of *Trichoderma* sp. [24], *Brevibacillus brevis* [25], and *Pantoea* sp. [25]. Argüello-Navarro and Moreno-Rozo [26] evaluated nitrogen-fixing bacteria isolated from cocoa-growing soils and found that strains of *Burkholderia* sp. and *Gluconacetobacter* sp. were most effective at increasing seedling growth.

Still, knowledge gaps remain as to the forms and rates of OM that should be applied, as well as the efficacy of different biofertilizers in optimizing both productivity and soil health in cocoa production systems. This study sought to evaluate the impacts of OM and EM on soil properties and cocoa yield over a period of fifteen months. It was hypothesized that both OM and EM would increase cocoa yield, albeit via different mechanisms—OM by increasing soil nutrient availability, as compost and vermicompost contain significant nutrient stocks that could be released through mineralization, and EM by increasing the abundance of beneficial microorganisms through the application of locally derived taxa selected for plant growth-promoting traits.

## 2. Materials and Methods

### 2.1. Site Description

The study was conducted on a cocoa farm in Guayaquil, Ecuador (2°23′25′′ S 80°13′02′ W), at an altitude of 12.01 m.a.s.l. According to the bioclimatic classification, this area corresponds to a semi-desert forest, characterized by two pronounced seasons: the rainy season, from December to May, and the dry season, which occurs from June to November. According to the meteorological station data at the site, the average annual precipitation at this site is 517.88 mm, and the average annual temperature ranges from 28.36 to 34.20 (daily high) and from 18.09 to 24.00 (daily low).

This area is characterized by regular topography, with minor slopes of up to 3%, leading to relatively homogeneous soils. According to soil maps of Ecuador, soils in the Cerecita Valley are characterized as Inceptisols and vary in texture from clay loam to clay. This is a healthy high-productivity farm without excessive incidence of soil-borne pathogens or heavy metals. As part of the integrated pest management strategy, fosetyl aluminum is applied annually at a rate of 2 kg/ha.

The trial includes two plots: Lot 1, with an area of 4 hectares, and Lot 22, covering 1.5 hectares. Both plots were established over 10 years ago with the cocoa clone CCN-51, planted in a double-row system at densities of 1480 trees/ha in Lot 1 and 1500 trees/ha in Lot 22.

Lot 1 follows a traditional full-sun design and is equipped with sprinkler irrigation. In contrast, Lot 22 features a mechanized layout, integrates naturally occurring forest trees at a density of 8 trees/ha, and uses a drip irrigation system.

### 2.2. Experimental Design

In May 2022, an experiment was established in randomized complete block design with three replicates to test two factors: OM amendment and inoculation with EM. These factors had three and two levels, respectively, for a total of six treatments. Three forms of OM were tested: compost, vermicompost, and cocoa leaf litter (the standard practice that served as control). Compost was derived from rice chaff, cow manure, and grass and was applied at a rate of 10 MT/ha/yr (Table 1). Vermicompost derived from the humus produced by California earthworms processing plant waste and applied at a rate of 10 MT/ha/yr. Neither amendment included living worms. The application rate of 10 MT/ha was based on standard recommendations for the rate to achieve an increase of 0.5% SOC over a three-year period. The litter treatment consisted of naturally fallen cocoa leaves mulched on the soil surface, whereas compost and vermicompost were applied as a surface layer around the trunk and mechanically incorporated into the soil to a depth of 0–15 cm (Table 1).

For the second treatment factor, a locally derived EM formulation was tested against uninoculated control. This formulation consisted of 13 different microorganisms, a mixture of local and external microorganisms. Prior to the trial, a screen was conducted to assess populations of microorganisms known in the literature to be of agricultural interest for their roles in organic matter decomposition, antagonism against pathogens, and capacity of establishing symbiosis with the root system. Given that all groups were found to be present, a decision was made to include them all to assess their effectiveness as inoculants and persistence in the soil. To determine the initial populations of these groups of soil microorganisms, a microbiological analysis was performed in the test areas using the microorganism plate counting methodology implemented by BioSeb Organics CIA. LTDA [27]. Based on this analysis, BioSeb Organics isolated eight types of microorganisms with the following methods.

*Lactobacillus* spp.: For selective isolation, serial dilutions of soil solution were made and inoculated onto petri plates containing MRS agar (De Man, Rogosa, Sharpe). The plates were incubated at 25 °C for 5 days to determine the number of colony-forming units per gram of soil. To identify the *Lactobacillus* genus, macroscopic and microscopic characterizations and physiological and biochemical tests were performed, yielding diverse representatives with specific characteristics that became part of a germplasm collection for research.

Actinomycetes: For the selective isolation of actinomycetes, a pretreatment was performed at a specific temperature and time (details are proprietary to BioSeb Organics), followed by serial dilutions that were inoculated onto petri plates containing glucose yeast malt (GYM) agar [28]. The plates were incubated at 25 °C for 15 days to determine the number of colony-forming units per gram of soil. To identify the actinomycetes, macroscopic and microscopic characterizations and physiological and biochemical tests were performed, allowing for diverse representatives with specific characteristics to be obtained, which became part of a germplasm collection for research.

Calcium-solubilizing bacteria: Serial dilutions were prepared and inoculated onto petri plates containing GYM agar. The plates were incubated at 25 °C for 8 days to determine the number of colony-forming units per gram of soil. Macroscopic and microscopic characterizations and physiological and biochemical tests were performed to identify calcium-solubilizing bacteria.

Phosphorus-solubilizing bacteria: Serial dilutions were prepared and inoculated onto petri plates containing Pikovskaya agar [29]. The plates were incubated at 25 °C for 8 days to determine the number of colony-forming units per gram of soil. To identify phosphorus-solubilizing bacteria, macroscopic and microscopic characterizations, as well as physiological and biochemical tests, were performed. In addition, the formation of a hydrolysis halo of the insoluble compound tricalcium phosphate was evaluated. For all types of microorganisms, once the strains were selected, bioaugmentation was performed in specific nutrient media, ensuring the concentration, viability, and purity of the bacterial consortium.

Potassium-solubilizing bacteria: Serial dilutions were made and inoculated onto monopetri plates containing Aleksandrov agar using potassium silicate as the inorganic source of potassium [30]. The plates were incubated at 25 °C for 5 days to determine the number of colony-forming units per gram of soil. Macroscopic and microscopic characterizations and physiological and biochemical tests were performed to identify potassium-solubilizing bacteria. In addition, the formation of a hydrolysis halo of the insoluble potassium compound was assessed.

Nitrogen-fixing bacteria (*Azotobacter* sp.): Serial dilutions were made and inoculated onto monopetri plates containing Ashby agar medium (Sigma-Aldrich, Saint Louis, MO, USA). The plates were incubated at 25 °C for 7 days to determine the number of colony-forming units per gram of soil. Macroscopic and microscopic characterization, as well as physiological and biochemical tests, were performed to identify *Azotobacter* sp. Once the microorganisms were selected, bioaugmentation was carried out in specific nutritional media, ensuring the concentration, viability, and purity of the bacterial consortium.

Nitrogen-fixing bacteria (*Azospirillum* sp.): Serial dilutions were made and inoculated onto monopetri plates containing Congo Red Malic Acid Agar medium [31]. The plates were incubated at 25 °C for 7 days to determine the number of colony-forming units per gram of soil. During this time, deep scarlet colonies were observed and isolated on nitrogen fixation biological (NFB) medium (Sigma-Aldrich, Saint Louis, MO, USA). This medium is frequently used to evaluate acetylene-reducing activity as an indicator of nitrogen fixation, which results in a blue to light green color change.

Macroscopic and microscopic characterizations, as well as physiological and biochemical tests, were performed to identify *Azospirillum* sp. Once the microorganisms were selected, they were bioaugmented in specific nutrient media, ensuring the concentration, viability, and purity of the bacterial consortium.

Yeasts: Serial dilutions were prepared and inoculated onto mono-Petri plates containing Sabouraud agar medium (Sigma-Aldrich, Saint Louis, MO, USA). The plates were incubated at 25 °C for 3 days to determine the number of colony-forming units per gram of soil. Yeast identification was performed through macroscopic and microscopic characterization, as well as physiological and biochemical tests. Once the microorganisms were selected, bioaugmentation was carried out in specific nutritional media, ensuring the concentration, viability, and purity of the bacterial consortium.

The five external microorganisms were part of the BioSeb Organics laboratory’s germplasm bank. They included *Bacillus* spp., *Pseudomonas* spp., mesophilic aerobes, and two fungi (*Trichoderma harzianum* and *T. viride*).

The microorganisms were applied as a combined cocktail in a 2% solution provided by BioSeb Organics. This solution was prepared by mixing 4 g each of *T. viride* and *T. harzanium* (each 3 × 10^10^ CFU/g), 1 L *Lactobacillus* sp. (4.3 × 10^9^ CFU/L), 1 L actinomycetes (1.89 × 10^9^ CFU/L), 1 L phosphorus-solubilizing bacteria (3 × 10^10^ CFU/L), 1 L calcium-solubilizing bacteria (1.26 × 10^11^ CFU/L), 1 L potassium-solubilizing bacteria (1.8 × 10^10^ CFU/L), 1 L *Azotobacter* sp. (7.5 × 10^10^ CFU/L), 1 L *Azospirillum* sp. (1.1 × 10^10^ CFU/L), 1 L yeasts (1.2 × 10^10^ CFU/L), 1 L *Bacillus* sp. (5 × 10^10^ CFU/L), 1 L *Pseudomonas* sp. (1.0 × 10^10^ CFU/L), and 1 L aerobic mesophylls (5 × 10^10^ CFU/L), then diluting that combined cocktail (2%) (*v*/*v*) in water. The solution was applied around the base of each tree with a backpack sprayer after clearing leaf litter from the soil surface. Each EM was applied with an initial dose of 2.0 L/ha, followed by a subsequent dose, each year, of 1.0 L/ha.

Management was the same for all lots except for the treatments applied.

### 2.3. Sample Collection and Processing

Microbiome sampling was conducted before and 15, 30, 45, and 60 days after the first application of EM. At each sampling date, ten trees were sampled per treatment. Using a soil corer, two soil samples were taken 30 cm from the base of the tree at a depth of 0–30 cm, and these samples did not include tree roots. After sampling, samples were preserved at a temperature of 4 to 8 °C until DNA extraction for amplicon sequencing.

Soil samples for chemical analysis were taken before the OM and EM applications. Again, ten trees were sampled per treatment. Using a soil corer, samples were taken 30 cm from the base of the tree at a depth of 0–30 cm.

Soil bulk density was measured using the cylinder method. Soil nutrient analysis was conducted at AgroAnálisis Laboratorios in Durán, Ecuador. Available micronutrients—Cu, Fe, Mn, and Zn—were extracted with DTPA + 0.01 M CaCl_2_ solution and quantified with atomic absorption. Available Ca, Mg, K, and Na were extracted with 1 M ammonium acetate (pH 7) and quantified with atomic absorption. P was extracted according to the Olsen method and quantified with UV-Vis spectrophotometry. Organic carbon was measured according to the Walkley–Black method. pH was measured in a 1:10 soil–water solution, and salinity was measured on a saturated paste using electrical conductance. Total N was measured with the Kjeldahl method.

Subsamples for plating were composited by OM treatment for each combination of lot, replicate, and date (*n* = 24). Pre-application counts were subtracted from post-application counts at each sampling location to account for natural differences in microbial abundance.

Every month, trees were evaluated for flowering index, cherelles, and pods. A flowering index was established to quantify the number of floral cushions of each cacao tree. In this index, 1 represents 0–100 floral cushions (low to normal flowering intensity), 2 represents 100–200 floral cushions, and 3 represents 200 or more floral cushions (maximum flowering intensity for healthy trees). Cherelles were defined as pods less than 2.5 cm in length. Big pods were defined as pods that would be harvested before the next sampling interval (i.e., within 15–30 days), length > 25 cm, and diameter > 10 cm. Ripe pods were defined as those ready for harvest and were identified by their yellow color.

### 2.4. Microbiome Sequencing and Data Processing

DNA was extracted from the soil samples described in Section 2.3 with a ZymoBIOMICS DNA Miniprep kit (Zymo Research, Irvine, CA, USA) and quantified with a Promega QuantiFluor^®^ system (Promega, Madison, WI, USA). Adapter sequences were ligated, amplicon length was verified on an agarose gel, and fragments were purified to remove free primers and dimers. Library preparation included the addition of Illumina Nextera XT index sequences (Illumina, San Diego, CA, USA), purification, and library normalization prior to sequencing. Paired-end amplicon sequencing (2 × 250 bp reads) was conducted for the 16S and ITS regions at a depth of 100,000 reads per sample on an Illumina MiSeq v3 platform. Prokaryotic communities were assessed by amplifying the V3–V4 regions of 16S rRNA using the primers 341F/805R [32], and fungal communities were assessed by amplifying the ITS region with the primers ITS86F/ITS4 [33,34]. The raw sequencing dataset has been deposited to the NCBI database under project number PRJNA1232705.

Primers were removed with the cutadapt tool [35], and reads were filtered, trimmed, and merged with the dada2 package v1.34 [36]. Taxonomy was assigned with the SILVA reference database v138.2 for prokaryotes and the UNITE database for fungi [37,38,39]. Samples were filtered to remove chloroplast and mitochondrial sequences, leaving 24,303 unique prokaryotic and 15,990 fungal amplicon sequence variants (ASVs).

Because sequencing depth can confound diversity analyses, two samples with fewer than 5000 reads were removed from both prokaryotic and fungal analyses, and analysis of variance (ANOVA) was used to test for differences in sequences per sample by EM and OM treatment on the remaining samples. Prior to computing alpha and beta diversity measures, all samples were rarefied to the minimum library size (11,212 sequences obtained from 16S primers, 5624 sequences obtained from ITS primers) using the vegan package v.2.6.4 [40]. Though less suitable for differential abundance analyses [41], rarefaction is a useful normalization method to account for differences in sequencing depth for alpha and beta diversity metrics [42] and performs similarly to more recently proposed normalization methods in ordination [43].

### 2.5. Data Analysis

Downstream data analysis was conducted in R software v4.4.2 [44]. Soil physical and chemical data were analyzed using a linear mixed model with EM, OM, and their interaction as fixed effects and lot as a random effect. ANOVA was used if the requirements of normality of residuals and homogeneity of variance were satisfied or could be satisfied with a transformation of the data; otherwise, a non-parametric Kruskal–Wallis test was implemented to test EM and OM effects individually. Effects were considered significant at *p* < 0.05, and post hoc Tukey HSD tests were used to assess differences among treatments with α = 0.05.

Microbial plate count data for the same sampling location before and after inoculation were compared using a non-parametric Kruskal–Wallis test at the same significance levels described above.

Three alpha (within-sample) diversity metrics (richness, Shannon index, Simpson index) were calculated and plotted with the vegan package. A generalized least squares linear model was constructed to test for differences in alpha diversity among EM and OM treatments or their interaction using the nlme package v.3.1.162 [45]. ANOVA was used if the requirements of normality of residuals and homogeneity of variance were satisfied or could be satisfied with a transformation of the data. Aligned ranks transformation (ART) ANOVA was used as an alternative test for the full model if the F values of ANOVAs on aligned responses not of interest were all zero; otherwise, Kruskal–Wallis tests were used to test the effects of EM and OM individually. To ensure that rarefaction had not altered the conclusions drawn, the tests were repeated with non-rarefied data.

Constrained analysis of principal coordinates (CAP) was used to ordinate prokaryotic and fungal communities separately. Rarefied ASV tables were transformed to relative abundance using the microbiome package v.1.28.0 [46], and Bray–Curtis dissimilarity matrices were ordinated with OM and EM treatments as fixed factors using the vegan package v.2.6-8 [40]. The effects of OM, EM, and their interaction were tested sequentially using permutational analysis of variance (PERMANOVA) with 5000 permutations.

Associations of microbial ASVs with OM and EM treatments were tested with linear models, as implemented in the MaAsLin2 package v.1.8.0 [47]. This approach was developed to detect statistically significant associations between multi-omics data and complex metadata. ASVs were required to be present in a minimum of 10% of samples to be included in the analysis. Generalized linear models were run on non-rarefied ASV tables subjected to compositional transformation independently for prokaryotes and fungi, with EM and OM treatments as fixed factors. These models yield coefficients (β) and significance statistics (q, p) for each ASV, with positive coefficients for a treatment indicating a higher relative abundance of that ASV in that treatment relative to the reference level. Associations were considered significant for this study with a Bonferroni–Holm-corrected q < 0.25 and *p* < 0.05.

Flowering index, cherelle, and pod data were fitted with non-zero-inflated negative binomial models suitable for count data with overdispersion. Because the same plants were assessed on each date, a repeated measures structure was used, with EM, OM, and their interaction as fixed effects, and lot and date as random effects. To determine whether the EM treatment effect was consistent over time, another negative binomial model was evaluated, with EM, date, and their interaction, as well as OM, as fixed effects, and lot as a random effect. Cumulative pod counts were compared using non-zero-inflated negative binomial models, with EM and OM as fixed effects, and lot as a random effect.

## 3. Results

### 3.1. Soil Physical and Chemical Properties

Only soil pH was affected by microbial amendment (Table 2). Application of EM lowered soil pH by 0.15 units (6.89 ± 0.05 vs. 6.94 ± 0.06, *p* < 0.05). Organic amendment treatment affected soil P (*p* < 0.001), K (*p* < 0.01), Na (*p* < 0.01), exchangeable sodium percentage (ESP) (*p* < 0.05), and Cu (*p* < 0.05) (Table 2). P tended to be ~20% higher in compost than vermicompost or litter, but the difference was not significant according to post hoc tests, and K was ~15% higher in compost than litter. Na and ESP were both ~30% higher in vermicompost than litter. Cu tended to be slightly higher in the compost treatment compared to vermicompost, but the post hoc test was only borderline significant (*p* = 0.059).

### 3.2. Soil Microbial Communities

#### 3.2.1. Culture-Dependent Quantification

The abundance of *Azotobacter* sp. and *Trichoderma* sp. was significantly increased by inoculation, and effects were borderline significant for the abundance of *Azospirillum* sp., Ca solubilizers, *Lactobacillus* sp., *Pseudomonas* sp., and P solubilizers (Figure 1). The abundance of *Azotobacter* sp. was ~50% higher, and the abundance of *Trichoderma* sp. was ~30% higher in the EM treatment (both *p* < 0.05). Of the aforementioned groups, *Pseudomonas* sp. were the only group whose abundance tended to decrease with inoculation (*p* = 0.061).

#### 3.2.2. Beta Diversity

Neither EM nor OM treatment had a significant effect on library size for prokaryotic or fungal communities (all *p* > 0.05). Constrained analysis of principal coordinates (CAP) showed very little influence of EM or OM on prokaryotic beta diversity (Figure 2A). PERMANOVA with 5000 permutations confirmed that neither factor had a significant effect (EM: R^2^ = 0.0052, F_1,136_ = 0.72, *p* > 0.05; OM: R^2^ = 0.014, F_2,136_ = 0.98, *p* > 0.05). Using non-rarefied data did not change the interpretation of the results for 16S communities (EM: R^2^ = 0.0052, F_1,136_ = 0.71, *p* > 0.05; OM: R^2^ = 0.014, F_2,136_ = 0.97, *p* > 0.05).

Organic amendment, but not microbial inoculation, accounted for a small but significant proportion of variation in fungal community composition (Figure 2B) (OM: R^2^ = 0.024, F_2,136_ = 1.66, *p* < 0.001; EM: R^2^ = 0.005, F_1,136_ = 0.75, *p* > 0.05). Again, using non-rarefied data did not change the interpretation of the results (OM: R^2^ = 0.023, F_2,136_ = 1.64, *p* < 0.01; EM: R^2^ = 0.005, F_1,136_ = 0.75, *p* > 0.05).

#### 3.2.3. Alpha Diversity

Neither microbial inoculation nor the form of the organic matter affected prokaryotic or fungal diversity, as measured by the Chao1, Shannon, or Simpson indices (all *p* > 0.05). A comparison of alpha diversity metrics using rarefied and non-rarefied sequencing data can be found in Table A1.

#### 3.2.4. Microbiome-Specific Linear Models

The relative abundance of fifteen prokaryotic ASVs was affected by OM treatment (Figure 3). Seven of those ASVs increased in relative abundance in the compost treatment relative to the litter control, including five members of the Bacillota and two members of the Pseudomonodota (Figure 3A). A strain of *Novibacillus thermophilus* was nearly twice as abundant in compost relative to litter (β = 1.90, *p* < 0.001). Only one prokaryotic ASV was decreased in the compost treatment relative to litter and was identified as a member of the order *Burkholderiales*. Six prokaryotic ASVs were relatively more abundant in the vermicompost treatment than the litter treatment, all belonging to the phylum Bacillota (Figure 3B). The ASV with the greatest increase in relative abundance in this treatment was identified as *Lysinibacillus* sp. (β = 1.77, *p* < 0.001). The same member of the order *Burkholderiales* that was decreased by compost application was also decreased by vermicompost application (β = 0.77, *p* < 0.001).

No prokaryotic ASVs were affected by EM application.

A total of 68 fungal ASVs responded to soil amendments, including 32 responsive to compost, 32 responsive to vermicompost, and 4 responsive to EM application (Figure 4). Of the fungi responsive to compost, the majority (twenty) belonged to the phylum Ascomycota, including eight members of the *Microascales*, the most-represented order. Twenty-four ASVs were more abundant in the compost treatment than litter, whereas eight were less abundant (Figure 4A).

Of the 32 fungal ASVs responsive to vermicompost, a similar number (21) belonged to the Ascomycota, but the ASVs were relatively evenly distributed among orders. Twenty-three ASVs were relatively more abundant in vermicompost than litter, whereas nine were less abundant (Figure 4B). The ASV with the highest coefficient, indicating the greatest increase in relative abundance in the vermicompost treatment, was identified as a member of the *Endogonomycetes* (genus incertae sedis) (β = 3.45, *p <* 0.001).

Four fungal ASVs responded to EM application, including one strain of *Trichoderma* sp. that increased in relative abundance (β = 1.07, *p* < 0.01) and three ASVs that decreased in relative abundance, including one *Cosmospora* sp. (β = −0.64, *p* < 0.01), one *Paracremonium* sp. (β = −0.90, *p* < 0.01), and one ASV that could not be identified, even to the phylum level (β = −1.07, *p* < 0.001).

### 3.3. Flowering, Fruit Set, and Yield

While flowering index was not affected by OM or EM applications (*p* > 0.05), monthly counts per tree of both cherelles and harvestable pods were higher in the EM treatment in both lots. Monthly cherelle formation was 19% higher in inoculated trees relative to uninoculated trees (EM: mean 1.46; Native: mean 1.22; *p* < 0.001). Similarly, inoculated trees formed 11% more harvestable pods (EM: mean, 1.30; Native: mean, 1.18; *p* < 0.001). Total annual cherelle production across both lots was 18% higher in inoculated trees (EM: mean 483; Native: mean 409; *p* < 0.01), and while total annual pod production tended to be 9% higher in inoculated trees, the effect was only borderline significant (EM: mean 430; Native: mean 394; *p* = 0.07). Evaluation of a model including the EM–date interaction showed that the interaction was significant in only two of the fifteen months evaluated (June and October 2024), with borderline significant effects in February, April, and August 2024.

Dry cocoa bean yields were not significantly different among treatments in either lot when considered over the full 18 months of the study (*p* > 0.05, Table 3).

## 4. Discussion

This study investigated the hypothesis that OM and locally derived EM application would increase cocoa yields via soil properties and microbial communities, respectively. In partial support of this hypothesis, compost had beneficial impacts on soil nutrients relevant to cocoa production, but without affecting pod or dry bean yields, while EM increased per-tree monthly production of cherelles by 19% and pods by 11%, with slight but significant impacts on microbial communities. These results support the complementarity of physicochemical and biological interventions to optimize soil health, as soil physicochemical properties change less rapidly than biological properties, but all are important over the long term.

Compost increased soil P and K by 15–20% relative to the litter-only control, indicating that it could be a reasonable source of these two critical nutrients for cocoa, though not a full replacement for other fertilizers. Vermicompost application, in contrast, only impacted soil Na and ESP. Given the high cost of this amendment relative to natural litterfall for no additional benefit and a slight increase in salinity, it does not appear that vermicompost application represents a worthwhile return on investment for this farm. Soil bulk density was constant across treatments, indicating that incorporation of compost and vermicompost with a motor hoe did not have lasting effects on porosity or soil compaction.

In addition to providing a short-term ancillary source of plant nutrients, increasing SOM stocks (and thus sequestering carbon) is a key reason to apply organic amendments. Here, a trend towards increased SOM was observed: compost and vermicompost treatments had ~4% more SOM than the litter-only control (1.19–1.20% vs. 1.05%), though the effect was not statistically significant. Soil organic matter and organic carbon (SOC) have been observed to shift only very slowly in tropical soils due to rapid turnover of SOM in these climates and unfavorable soil geochemical properties for C stabilization [48]. For example, SOC increased at a rate of only 7–9‰ per year in cocoa agroforestry systems in Cameroon [49], and even 15 years of experimentally doubled litter additions, did not increase SOC in another tropical forest soil [48]. Given the promising trend towards increased SOM even after 15 months in the present study, it is reasonable to assume that applying these amendments over a longer duration would compound these effects, with corresponding benefits for soil health and carbon sequestration. Future studies should also explicitly investigate whether application of these treatments increased rates of litter decomposition, which could release more plant-available nutrients.

pH was the only soil physicochemical property affected by EM application, which lowered soil pH by 0.15 units, i.e., from 6.94 to 6.89 (Table 2). This effect may be due to the inclusion of *Lactobacillus* sp., which produces lactic acid, or the secretion of a number of other organic acids known to be produced by rhizosphere microorganisms. The magnitude of this impact is slight, and the soil pH remained in an optimal range for cocoa, which can tolerate soil pH as low as 5.5 [50], but effects could be even greater at the rhizoplane where microorganisms are concentrated. It is unsurprising that few bulk soil properties were affected by the application of a relatively small volume of biofertilizer that did not contain additional plant nutrients. In addition, impacts of the inoculated microorganisms would likely be greater in the rhizosphere, where microorganisms congregate due to resource availability, than in the bulk soil that was sampled for soil analyses.

The impacts of OM outweighed the impacts of EM application on microbial communities as a whole. This outcome is reasonable in light of the fact that compost and vermicompost provide novel and varied substrates for microbial metabolism relative to cocoa leaf litter alone, whereas inoculated strains must compete with the native microbial community to establish. Compost and vermicompost application shifted fungal community composition and altered the relative abundance of 15 prokaryotic and 64 fungal ASVs, as compared to 4 ASVs whose relative abundance was impacted by biofertilizer application. Culture-dependent and culture-independent methods provided complementary information in this study. Plate counts targeting specific groups picked up changes in total abundance of genera and functional groups hypothesized to be important for plant growth promotion, while amplicon sequencing identified ASV-level shifts in relative abundance that could indicate key taxa linking soil management and plant outcomes.

Vuolo et al. [51] suggest three potential outcomes when biofertilizers are applied: native communities outcompete the introduced strains, resulting in no change; the inoculated strains establish, permanently shifting community composition; and inoculated strains cause a transient shift, but the original composition is ultimately restored. Culture-dependent methods showed a persistent increase in *Azotobacter* sp., which was included in the microbial consortium, in soils where EM was applied (Figure 1). While research specifically relevant to growth promotion mechanisms in cocoa is lacking, this genus has been found in the cocoa rhizosphere [52] and is widely used in agricultural biofertilizers because it includes members capable of free-living N fixation, P solubilization, and plant growth stimulation [51]. No signs of yield-limiting nitrogen deficiency were observed in these cocoa trees, but *Azotobacter* sp. could also have increased cocoa pod production via production of secondary metabolites, phytohormones, or siderophores [53]. A significant increase in the total abundance of *Trichoderma* sp. was likewise observed after inoculation, which is especially noteworthy given the body of research on this genus as a potential plant-growth promoter and biocontrol agent in cocoa [24,54,55,56,57]. Other studies have shown that *Trichoderma* sp. colonizes cacao endophytically [58,59] and directly improves cacao seedling growth (particularly root growth) and drought tolerance [18,24], as well as suppressing pathogens [54,57,60]. Though explicitly testing the mechanisms of growth promotion was beyond the scope of this study, isolating the taxa whose abundance increased following inoculation and sequencing their genomes would be a useful next step.

Given the small sample size for the culture-dependent methods in the present study, borderline significant impacts on microbial communities are also worth discussing. *Lactobacillus* sp., *Azospirillum* sp., Ca solubilizers, and P solubilizers tended to increase in the EM treatment, with potential ramifications for plant nutrition and increased pod set. *Lactobacillus* sp. are involved in cocoa bean fermentation but not well established as plant growth promoters. In light of evidence that rhizosphere acidification caused by their lactic acid production could increase plant cadmium uptake [61], this genus may only be suitable as a biofertilizer for cacao in soils where cadmium is not a concern. *Azospirillum* sp. are diazotrophic and often solubilize phosphate and produce phytohormones [51]. Ca and P solubilizers may be especially useful for meeting tree nutrient demand without additional fertilizer inputs in cocoa systems. In these systems, nutrient stocks in litterfall can often be comparable to the amounts of nutrients exported in harvest, but the release of Ca and especially P bound in organic forms is slow [50]. Interestingly, the total abundance of *Pseudomonas* sp., a large genus shown to be beneficial in plants including cacao [62,63], decreased after biofertilizer application. Possible explanations include competitive exclusion or direct inhibition by introduced microorganisms. The culture-dependent methods in this study required compositing samples by EM treatment, leading to a small sample size, and these results should thus ideally be validated in larger follow-up experiments.

Amplicon sequencing found no effect of EM application on microbial diversity and only a very limited effect on fungal community composition as a whole but highlighted specific biofertilizer-responsive ASVs. The lack of effect on bacterial or fungal diversity is consistent with a 2024 meta-analysis of 335 studies on the impacts of biofertilizers, which found that inoculants did not alter diversity but could have profound impacts on community composition [64]. Cornell et al. [65] propose a hypothesis that might explain the slight but significant effect on fungal but not bacterial community composition that was observed here: the lower diversity of fungal communities may render them more susceptible to invasion; whereas greater dispersal of bacteria could create communities where the invaders’ target niche is already occupied by taxa with similar ecological functions.

One strain of *Trichoderma* sp. increased in relative abundance following EM application. It is plausible that this is the same strain that was inoculated and that increased in absolute abundance according to culture-dependent methods, although it is not possible to confirm this because neither the original EM nor the recovered isolates were sequenced. Regardless, an increase in any members of this genus is likely beneficial for cacao growth and biocontrol of pathogens. Limited research on the cacao microbiome, combined with the use of amplicon sequencing alone and the inability to identify more than a few of the treatment-responsive ASVs in this study to genus or species level, makes it difficult to propose mechanistic hypotheses about the impact of the other taxa on cacao here. Taxonomic identification was constrained by the limited coverage of soil microorganisms, especially those from tropical soils, in the reference databases used to assign taxonomy. Only two of the ASVs responsive to OM were also found in the 137 cacao endophytes recently isolated and identified [66], a *Fusarium* sp. and *Fusarium oxysporum* whose relative abundance was increased by vermicompost application. Two previous studies of cacao endophytes also noted that *Fusarium oxysporum* was present, but without clarifying its ecological role [67,68]. Additional studies integrating other omics methods and functional assays, such as those conducted in other systems [69,70], would have provided interesting and useful insight into the responsive taxa, but these were beyond the scope of the present study. Despite the inability of this study design to elucidate the mechanisms involved in improving agronomic outcomes, biofertilizer application appears to have impacted cocoa productivity without impacting microbial community composition overall, indicating that even shifts in a small number of groups or ASVs can be functionally relevant.

Trees receiving EM converted a higher proportion of flowers to cherelles and harvestable pods each month in both lots, although only the effect on cherelles translated to differences in annual totals (Figure 5). While flowering intensity is regulated in cocoa by factors, including the number of pods already present, the production of harvestable pods is limited by assimilate production, and the tree aborts immature pods at the cherelle stage (“cherelle wilt”) to control the number of pods [71]. Cherelle wilt is thus visible when a large number of the immature pods forming after pollination fail to develop further, leaving only the pods that can be supported by available resources. Inoculation resulted in an increased abundance of microorganisms known to fix N (e.g., *Azotobacter*, *Azospirillum*) and solubilize P and Ca, which may have contributed to plant nutritional status and thus alleviated the primary constraint on assimilate production. The effect size (9–19%) is large, especially given recent work showing the inconsistent results and poor efficacy of commercial biostimulants [14,72,73]. This increase in monthly pod production from 1.18 to 1.30 pods per tree would be worth over 1500 USD/ha at current cocoa prices, assuming a planting density of 1500 trees/ha and using the CCN51 pod index of 15.2 [74]. However, the borderline significant effect on annual totals suggests that further research is needed on the mechanisms involved and the variability of biofertilizer response between individual trees and over time.

Although cumulative annual pod production was only borderline significant (*p* = 0.07), the inoculated trees consistently produced more cherelles and harvestable pods during specific months outside the traditional peak harvest period of November to January. Significant effects were observed in seasonality in months with different climates and phenologies, i.e., June (a wet month of peak flowering) and October (a drier month with peak juvenile pods), with trends toward significance in February, April, and August. These months are characterized by environmental differences but are similar as periods of naturally lower productivity of maturity pods, which may be shaped by internal resource allocation dynamics or phenological cycles. The observed treatment effects during these off-peak months suggest that biofertilizer inoculation may enhance reproductive consistency, potentially by improving resource availability at times when cacao trees are less likely to invest in fruit production.

Importantly, the lack of treatment effect during peak productivity months, which are likely driven by other dominant agronomic or climatic factors (e.g., photoperiod, carbohydrate availability, or canopy status), may explain why cumulative yield was not significantly higher. However, the gain in off-peak yield is agronomically relevant, offering growers a more distributed harvest curve, which can reduce labor bottlenecks, increase pod availability for fermentation continuity, and improve year-round continuity of farm operations. Notably, the effect was observed in both traditional and mechanized cultivation, indicating that EM could be relevant to diverse cocoa production systems. 

## 5. Conclusions

Whether microbial inoculants such as this one gain traction depends, in part, on broader economic and environmental considerations such as labor requirements and availability, fluctuating cocoa prices, and the long-term impacts of microbial biofertilizers, all of which require further research. This result supports a shift towards a model of biofertilizer development similar to the method used here, in which microorganisms isolated from a specific farm undergo trait validation in a nearby laboratory and promising strains are propagated and returned to the same farm, rather than relying on commercial products purported to be applicable across geographies. Such locally derived biofertilizers could be combined with organic amendments such as the compost tested in this study to optimize pod yields and soil health on cocoa farms.

## 6. Patents

A provisional patent has been filed for the microbial inoculant used in this study.

## Figures and Tables

**Figure 1 microorganisms-13-01408-f001:**
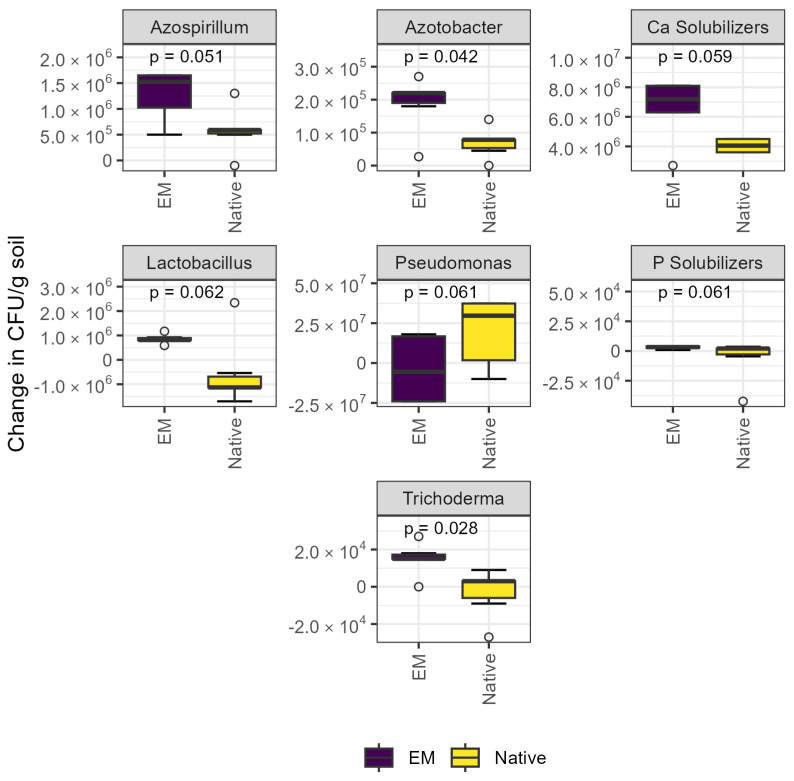
Culture-dependent methods revealed abundance of selected microbial groups significantly increased after inoculation. Plate counts revealed that inoculation increased the abundance of *Azotobacter* and *Trichoderma* sp. *(p* < 0.05) and tended to affect the abundance of *Azospirillum* sp., *Lactobacillus* sp., *Pseudomonas* sp., and Ca and P solubilizers. CFU: colony-forming units; EM: efficient microorganisms.

**Figure 2 microorganisms-13-01408-f002:**
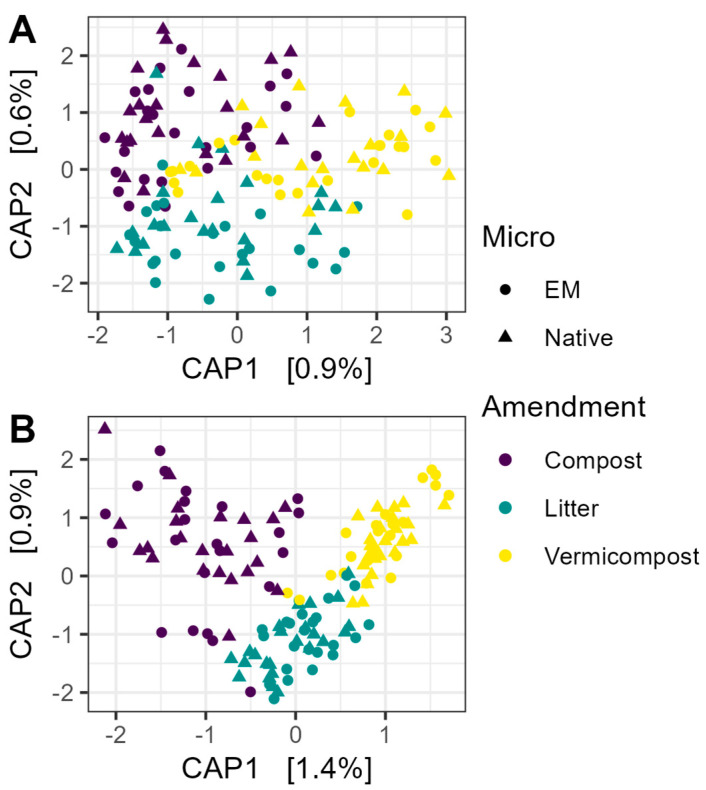
Constrained analysis of principal coordinates (CAP) of microbial communities. (**A**) Beta diversity of prokaryotic communities was not explained by microbial inoculation, nor organic matter amendment. (**B**) Organic amendment had a small but significant effect on fungal community composition (R^2^ = 0.024; F_2,136_ = 1.66, *p* < 0.001). The ordination shown was based on Bray–Curtis dissimilarity matrices constructed from rarefied data, and significance was tested using PERMANOVA with 5000 permutations. EM: efficient microorganisms.

**Figure 3 microorganisms-13-01408-f003:**
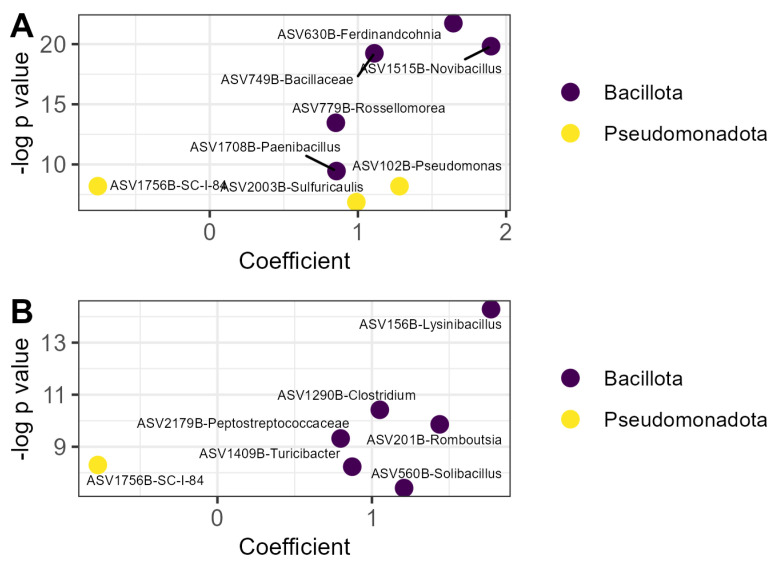
Prokaryotic ASVs responsive to organic matter amendment: (**A**) ASVs whose relative abundance was affected by compost; (**B**) ASVs whose relative abundance was affected by vermicompost. Coefficients are derived from the microbiome-specific linear models and indicate the magnitude and direction of the change in relative abundance in that treatment relative to the litter control, while *p* values indicate the significance of the model for that ASV. ASVs are labeled with the lowest taxonomic level to which they could be identified with the SILVA reference database v138.2 and colored according to phylum. ASV: amplicon sequence variant.

**Figure 4 microorganisms-13-01408-f004:**
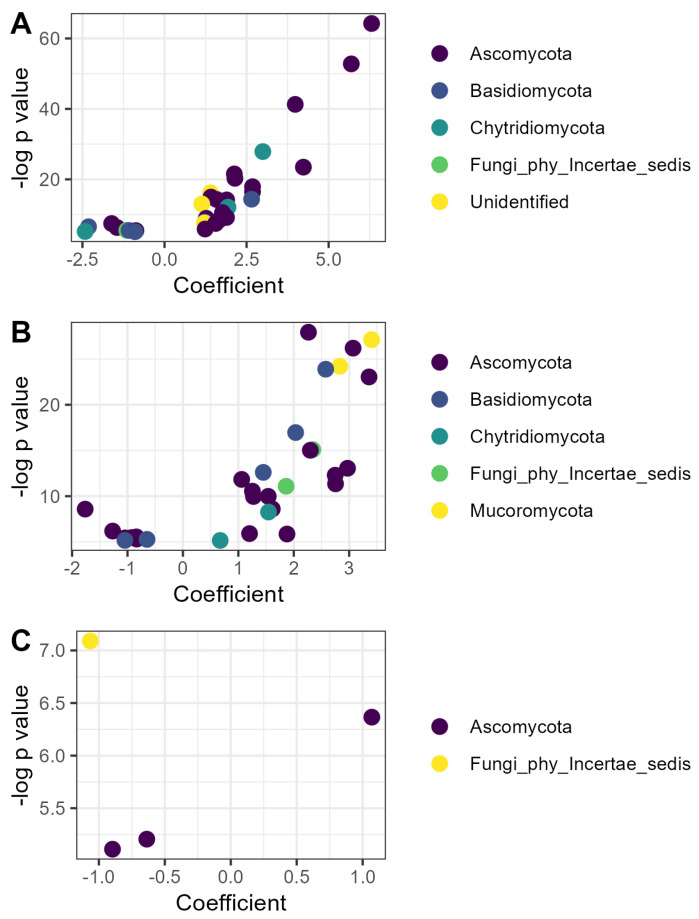
Fungal ASVs responsive to organic matter amendment or microbial inoculation: (**A**) ASVs whose relative abundance was affected by compost; (**B**) ASVs whose relative abundance was affected by vermicompost; (**C**) ASVs whose relative abundance was affected by inoculation with efficient microorganisms. Coefficients are derived from the microbiome-specific linear models and indicate the magnitude and direction of the change in relative abundance in that treatment relative to the litter control, while *p* values indicate the significance of the model for that ASV. ASVs are colored according to phylum as identified with the SILVA reference database v.138.2.

**Figure 5 microorganisms-13-01408-f005:**
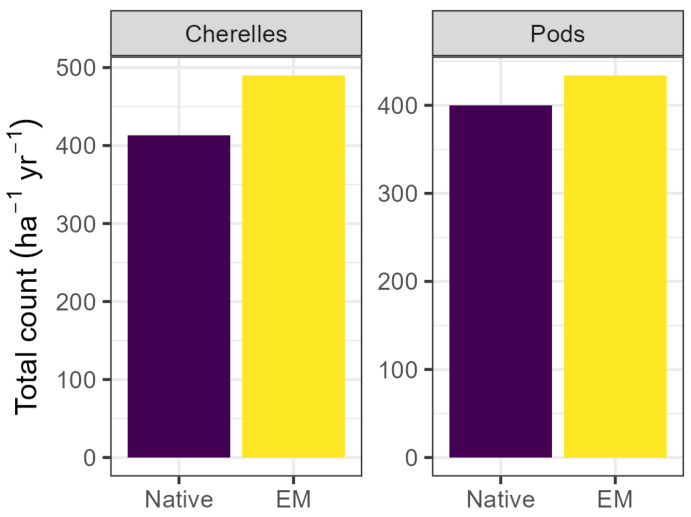
Cumulative annual cherelle and pod counts per hectare. Total cherelles increased by 18% via EM application (*p* < 0.01), and total pods tended to increase but the effect was only borderline significant (*p* = 0.07). Values represent the sum of both lots and are reported per ha per year. EM: efficient microorganisms.

**Table 1 microorganisms-13-01408-t001:** Nutrient analysis and application methods for organic matter treatments.

	Compost	Vermicompost	Litter
Organic matter (%)	48.09	31.57	85.41
Moisture content (%)	38.63	35.13	12.94
N (%)	1.63	1.68	1.6
P (%)	1.47	Trace	Trace
K (%)	2.27	1.1	0.38
Ca (%)	3.09	2.75	3.86
Application rate	10 MT/ha/yr	10 MT/ha/yr	1.6 MT/ha/yr
Incorporation method	Rototiller	Rototiller	Mulched on surface

**Table 2 microorganisms-13-01408-t002:** Soil physical and chemical properties according to treatment (mean ± standard error). Because the EM–OM interaction was not significant in any case, post hoc comparisons were conducted for EM and OM separately. Values followed by the same letter are not significantly different at α = 0.05. BD: bulk density; CEC: cation exchange capacity; EC: electrical conductivity; ESP: exchangeable sodium percentage; SOC: soil organic carbon.

	EM		OM		
	EM	Native	Compost	Litter	Vermicompost
BD (g/cm^3^)	1.46 ± 0.01 a	1.46 ± 0.01 a	1.46 ± 0.02 a	1.46 ± 0.02 a	1.47 ± 0.02 a
Ca (meq/100 g)	24.11 ± 0.37 a	24.29 ± 0.37 a	24.00 ± 0.45 a	24.46 ± 0.45 a	24.15 ± 0.46 a
CEC (meq/100 g)	32.36 ± 0.59 a	32.77 ± 0.70 a	31.91 ± 0.83 a	32.95 ± 0.78 a	32.84 ± 0.76 a
Cu (ppm)	10.84 ± 0.77 a	11.44 ± 0.89 a	12.99 ± 1.18 a	10.72 ± 0.95 a	9.71 ± 0.82 a
EC (dS/m)	0.83 ± 0.05 a	0.72 ± 0.04 a	0.79 ± 0.06 a	0.81 ± 0.05 a	0.73 ± 0.05 a
ESP (%)	0.96 ± 0.05 a	1.06 ± 0.07 a	1.00 ± 0.08 ab	0.88 ± 0.06 b	1.15 ± 0.07 a
Fe (ppm)	152.90 ± 15.66 a	148.50 ± 12.90 a	140.40 ± 16.2 a	161.53 ± 21.04 a	150.17 ± 14.99 a
K (meq/100 g)	1.20 ± 0.03 a	1.25 ± 0.04 a	1.32 ± 0.05 a	1.15 ± 0.03 b	1.20 ± 0.04 ab
Mg (meq/100 g)	5.01 ± 0.19 a	5.31 ± 0.23 a	5.12 ± 0.23 a	5.18 ± 0.28 a	5.18 ± 0.26 a
Mn (ppm)	21.05 ± 2.40 a	27.47 ± 3.96 a	21.74 ± 3.71 a	26.34 ± 4.69 a	24.68 ± 3.75 a
Na (meq/100 g)	0.31 ± 0.02 a	0.34 ± 0.02 a	0.32 ± 0.03 ab	0.29 ± 0.02 b	0.38 ± 0.02 a
P (ppm)	41.13 ± 3.42 a	43.52 ± 3.94 a	49.82 ± 3.54 a	38.86 ± 5.05 b	38.30 ± 4.53 b
pH	6.79 ± 0.05 b	6.94 ± 0.06 a	6.92 ± 0.05 a	6.87 ± 0.07 a	6.80 ± 0.08 a
SOC (%)	1.19 ± 0.03 a	1.17 ± 0.02 a	1.19 ± 0.03 a	1.15 ± 0.03 a	1.20 ± 0.03 a
Zn (ppm)	12.47 ± 1.10 a	14.35 ± 1.16 a	12.09 ± 1.42 a	14.27 ± 1.48 a	13.88 ± 1.29 a

**Table 3 microorganisms-13-01408-t003:** Comparison of annual dry bean yield (mean ± standard deviation) per treatment and lot over the 18-month duration of the study. Two lots were sampled in this study: Lot 1, which is 4 ha in size with a traditional full-sun design; and Lot 22, which is 1.5 ha with a mechanized design and includes naturally occurring forest trees. ANOVA showed that annual yield was not significantly different among treatments in either lot (*p* > 0.05).

Treatment	Lot	Yield (t/ha/yr)
Compost + EM	1	1.88 ± 0.36
Compost + Native		1.92 ± 0.46
Litter + EM		2.26 ± 0.57
Litter + Native		1.77 ± 0.13
Vermicompost + EM		1.78 ± 0.35
Vermicompost + Native		1.94 ± 0.21
Compost + EM	22	2.39 ± 0.06
Compost + Native		2.48 ± 0.10
Litter + EM		2.48 ± 0.22
Litter + Native		2.56 ± 0.16
Vermicompost + EM		2.25 ± 0.51
Vermicompost + Native		2.61 ± 0.37

## Data Availability

The sequencing data originating from this study has been made publicly available in the NCBI database under project number PRJNA1232705.

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
