# Peer review of "Optimizing Cocoa Productivity Through Soil Health and Microbiome Enhancement: Insights from Organic Amendments and a Locally Derived Biofertilizer"

_microorganisms, 2025, doi:10.3390/microorganisms13061408_

Round 1

Reviewer 1 Report

Comments and Suggestions for Authors

This manuscript presents a compelling study on enhancing cocoa productivity in Ecuador through organic amendments and a locally-derived biofertilizer. The research addresses a critical need in sustainable cocoa farming by investigating the potential of microbiome management, an area that remains relatively unexplored in tropical agroecosystems. The study’s strength lies in its multi-faceted approach, integrating soil chemical analysis, both culture-dependent and independent microbial community assessments, and yield data to provide a holistic understanding of the system.

The experimental design, a randomized complete block with multiple replicates, ensures statistical rigor, while the 15-month monitoring period offers valuable insights into the long-term effects of the treatments on cocoa productivity. The comprehensive methodology, employing both plate counts and amplicon sequencing (16S and ITS), strengthens the microbial analysis. Furthermore, the thorough assessment of soil physicochemical properties, microbial diversity, and cocoa yield metrics contributes to the robustness of the findings. The study’s practical implications are significant, suggesting that biofertilizers can improve cocoa yields (11–19% increase in pod production) without requiring extensive soil amendments, offering potentially cost-effective solutions for farmers.

However, the study also exhibits some limitations. A major challenge lies in the limited taxonomic resolution of the sequencing data, especially concerning treatment-responsive ASVs. The fact that many of these variants could not be confidently assigned to specific genera or species hampers the ability to formulate precise mechanistic hypotheses on how microbial changes influence cacao growth. The absence of complementary functional analyses, such as metagenomics or functional gene profiling, restricts insights into microbial metabolic contributions, like nitrogen fixation or phosphate solubilization, that could underpin the observed yield improvements. Future research should supplement these results, as seen in Fallah et al. (2023) and Liu et al. (2021), where integrated multi-omics approaches revealed key microbial pathways driving crop benefits.”

Fallah N, et al. Sustained organic amendments utilization enhances ratoon crop growth and soil quality by enriching beneficial metabolites and suppressing pathogenic bacteria[J]. Frontiers in Plant Science, 2023, 14, 1273546.

Liu Q, et al. Bio-fertilizer Affects Structural Dynamics, Function, and Network Patterns of the Sugarcane Rhizospheric Microbiota[J]. Microbial Ecology, 2021, 84, 1195-1211.

Another area where the manuscript could be strengthened is in the mechanistic interpretation of the microbial shifts. While taxa such as Azotobacter and Trichoderma are mentioned, the discussion does not delve deeply into how these microbes might directly or indirectly enhance cocoa productivity. Clarifying these functional roles would greatly improve the readers’ understanding of the biological processes at play.

From a statistical perspective, some results are somewhat borderline—particularly the p-value of 0.07 for annual pod totals—which weakens the strength of the conclusions drawn. Additionally, the cost-benefit analysis presented focuses primarily on per-hectare treatment costs and yields, but it omits broader economic considerations such as labor input, market fluctuations, and potential long-term environmental impacts. These factors are crucial for a comprehensive assessment of the economic viability and practicality of adopting these treatments on a larger scale.

Finally, while the seasonal variability observed—significant yield effects in only two months and borderline effects in others—acknowledges potential seasonality, the study does not thoroughly explore the reasons behind this pattern. Understanding environmental or management factors driving seasonal differences would enhance the applicability of the findings for farmers seeking reliable year-round productivity improvements.

Author Response

This manuscript presents a compelling study on enhancing cocoa productivity in Ecuador through organic amendments and a locally-derived biofertilizer. The research addresses a critical need in sustainable cocoa farming by investigating the potential of microbiome management, an area that remains relatively unexplored in tropical agroecosystems. The study’s strength lies in its multi-faceted approach, integrating soil chemical analysis, both culture-dependent and independent microbial community assessments, and yield data to provide a holistic understanding of the system.

Thank you.

The experimental design, a randomized complete block with multiple replicates, ensures statistical rigor, while the 15-month monitoring period offers valuable insights into the long-term effects of the treatments on cocoa productivity. The comprehensive methodology, employing both plate counts and amplicon sequencing (16S and ITS), strengthens the microbial analysis. Furthermore, the thorough assessment of soil physicochemical properties, microbial diversity, and cocoa yield metrics contributes to the robustness of the findings. The study’s practical implications are significant, suggesting that biofertilizers can improve cocoa yields (11–19% increase in pod production) without requiring extensive soil amendments, offering potentially cost-effective solutions for farmers.

However, the study also exhibits some limitations. A major challenge lies in the limited taxonomic resolution of the sequencing data, especially concerning treatment-responsive ASVs. The fact that many of these variants could not be confidently assigned to specific genera or species hampers the ability to formulate precise mechanistic hypotheses on how microbial changes influence cacao growth. The absence of complementary functional analyses, such as metagenomics or functional gene profiling, restricts insights into microbial metabolic contributions, like nitrogen fixation or phosphate solubilization, that could underpin the observed yield improvements. Future research should supplement these results, as seen in Fallah et al. (2023) and Liu et al. (2021), where integrated multi-omics approaches revealed key microbial pathways driving crop benefits.”

Fallah N, et al. Sustained organic amendments utilization enhances ratoon crop growth and soil quality by enriching beneficial metabolites and suppressing pathogenic bacteria[J]. Frontiers in Plant Science, 2023, 14, 1273546.

Liu Q, et al. Bio-fertilizer Affects Structural Dynamics, Function, and Network Patterns of the Sugarcane Rhizospheric Microbiota[J]. Microbial Ecology, 2021, 84, 1195-1211.

Thank you – we agree that inability to completely identify taxa to species level and use of amplicon sequencing alone are limitations of the study and have added an acknowledgement of these limitations to the discussion (L593-598). Taxonomic identification was constrained by the limited coverage of soil microorganisms, especially those from tropical soils, in the reference databases used to assign taxonomy. Given this limitation, we chose not to speculate on mechanistic hypotheses that would require a more precise knowledge of the taxa involved and their genomes. We also agree that additional ‘omics methods and functional assays would have provided interesting and useful insight into the capabilities of these taxa, but these were beyond the scope of the study. We have added a brief mention of the utility of such complementary measures as future extensions of this work and included the citations you provided (L603-607).

Another area where the manuscript could be strengthened is in the mechanistic interpretation of the microbial shifts. While taxa such as Azotobacter and Trichoderma are mentioned, the discussion does not delve deeply into how these microbes might directly or indirectly enhance cocoa productivity. Clarifying these functional roles would greatly improve the readers’ understanding of the biological processes at play.

While research specifically on cocoa is limited, especially for Azotobacter sp., we have expanded the discussion to include more of the relevant literature on these taxa (L545-557). This section now describes the most plausible mechanism of growth promotion for Azotobacter sp., namely nitrogen fixation, as well as other growth-promoting capabilities described in the literature, and reviews evidence from other Trichoderma studies in cacao to propose other plausible mechanisms. We also suggest next steps to explore the mechanisms involved in future studies (L557-559).

From a statistical perspective, some results are somewhat borderline—particularly the p-value of 0.07 for annual pod totals—which weakens the strength of the conclusions drawn. Additionally, the cost-benefit analysis presented focuses primarily on per-hectare treatment costs and yields, but it omits broader economic considerations such as labor input, market fluctuations, and potential long-term environmental impacts. These factors are crucial for a comprehensive assessment of the economic viability and practicality of adopting these treatments on a larger scale.

We agree that the borderline p value for annual pod totals means that we cannot make strong claims about the effects of inoculation on farm-level yields, and as we discuss, this borderline result “suggests that further research is needed on variability of biofertilizer response between individual trees and over time” (L624-725). We have added two paragraphs exploring this result further, specifically the seasonality of effects since we observed significant monthly but not annual yield impacts (L626-643). We have remained conservative in the language used to describe statistically borderline results.

We also agree that the broader economic considerations mentioned would affect adoption of any new management practices. We have added a sentence to the conclusion to this effect: “Whether microbial inoculants such as this one gain traction depends in part on broader economic and environmental considerations such as labor requirements and availability, fluctuating cocoa prices, and long-term impacts of microbial biofertilizers, all of which require further research” (L659-662). Given that the broader economic perspectives are outside our areas of expertise, we removed the "costs" section from the results and table and focused only on the agronomic and microbiological outcomes.

Finally, while the seasonal variability observed—significant yield effects in only two months and borderline effects in others—acknowledges potential seasonality, the study does not thoroughly explore the reasons behind this pattern. Understanding environmental or management factors driving seasonal differences would enhance the applicability of the findings for farmers seeking reliable year-round productivity improvements.

Thank you for this observation. We have added two paragraphs of some of the potential drivers of the observed seasonality, including plant phenology and precipitation patterns (L628-646).

Reviewer 2 Report

Comments and Suggestions for Authors

This study investigates the effects of organic matter (OM) amendments (compost, vermicompost, and litter) and efficient microorganisms (EM), as biofertilizers, on cocoa productivity and soil health at a farm in Guayaquil, Ecuador. The authors assessed chemical, physical, and microbial community changes in soil, along with cocoa flowering, fruit setting (cherelle formation), and yields over a 15-month period. Culture-dependent and independent methods were employed to characterize microbial communities. Results showed that compost increased soil nutrients (particularly P and K), vermicompost raised soil sodium levels, and EM slightly decreased soil pH and enhanced beneficial microbial groups such as Azotobacter and Trichoderma. Cherelle formation and pod yield increased notably with EM, while OM amendments mostly influenced microbial composition without significant yield improvements.

General Comments:

The manuscript addresses an important topic of sustainable agriculture and provides relevant insights into integrated soil health and microbial management strategies in cocoa cultivation. The experimental design and methods are largely appropriate, rigorous, and clearly described. The novelty is apparent in exploring the combination of organic amendments and microbial inoculants on cocoa productivity and soil properties. However, some methodological clarifications, deeper interpretation of microbial community results, and better integration of microbial and agronomic data would strengthen the manuscript. The discussion is comprehensive but occasionally speculative and could be more clearly linked to results.

Overall, the manuscript is of high scientific quality and relevance but requires minor revisions to enhance clarity and depth of interpretation.

Specific Comments

Lines 92-98: Clarify your expected mechanisms more explicitly—why EM is specifically expected to improve yields through beneficial microbial populations, and why OM amendments are expected to enhance yields through nutrient availability. This will help link your hypothesis directly to your results and discussion.

Line 105: Clarify if the given temperature range is daily average, daily max-min, or seasonal ranges.

Lines 120-126: Confirm explicitly if the application rates for compost and vermicompost were based on previous research or standard recommendations.

Lines 133-195: Briefly justify why these particular groups of microorganisms were selected for inclusion in the EM formulation. Were there functional assays conducted prior to selecting these microorganisms?

Lines 279-355: Given that microbial beta diversity effects are minimal, provide a concise justification or hypothesis about why significant agronomic outcomes (pod yields) occurred despite the small effects on microbial community composition overall.

Lines 433-565: Improve integration of microbial data and agronomic outcomes. Currently, the discussion about specific microbial ASVs (e.g., Trichoderma spp.) and their possible roles in plant growth and productivity could be expanded with stronger references to existing cocoa-specific microbial research.

Line 446: Discuss briefly whether the observed incremental increase in SOM might have greater significance or cumulative effects in the longer-term management scenarios.

Line 543: A brief explanation or definition of "cherelle wilt" would enhance reader understanding of pod production constraints.

Author Response

This study investigates the effects of organic matter (OM) amendments (compost, vermicompost, and litter) and efficient microorganisms (EM), as biofertilizers, on cocoa productivity and soil health at a farm in Guayaquil, Ecuador. The authors assessed chemical, physical, and microbial community changes in soil, along with cocoa flowering, fruit setting (cherelle formation), and yields over a 15-month period. Culture-dependent and independent methods were employed to characterize microbial communities. Results showed that compost increased soil nutrients (particularly P and K), vermicompost raised soil sodium levels, and EM slightly decreased soil pH and enhanced beneficial microbial groups such as Azotobacter and Trichoderma. Cherelle formation and pod yield increased notably with EM, while OM amendments mostly influenced microbial composition without significant yield improvements.

General Comments:

The manuscript addresses an important topic of sustainable agriculture and provides relevant insights into integrated soil health and microbial management strategies in cocoa cultivation. The experimental design and methods are largely appropriate, rigorous, and clearly described. The novelty is apparent in exploring the combination of organic amendments and microbial inoculants on cocoa productivity and soil properties. However, some methodological clarifications, deeper interpretation of microbial community results, and better integration of microbial and agronomic data would strengthen the manuscript. The discussion is comprehensive but occasionally speculative and could be more clearly linked to results.

Overall, the manuscript is of high scientific quality and relevance but requires minor revisions to enhance clarity and depth of interpretation.

 Thank you for your positive comments. We have made extensive revisions to the manuscript in accordance with the comments of all three reviewers, including clarifying the methods and adding depth to our interpretation of both microbial and agronomic data. Please see specific point-by-point responses to all reviewers below.

Specific Comments

Lines 92-98: Clarify your expected mechanisms more explicitly—why EM is specifically expected to improve yields through beneficial microbial populations, and why OM amendments are expected to enhance yields through nutrient availability. This will help link your hypothesis directly to your results and discussion.

Thank you for the suggestion. We have revised our hypothesis to include more information about the expected mechanism: “It was hypothesized that both OM and EM would increase cocoa yield, albeit via different mechanisms: OM by increasing soil nutrient availability, as compost and vermicompost contain significant nutrient stocks that could be released through mineralization, and EM by increasing the abundance of beneficial microorganisms through application of locally-derived taxa selected for plant-growth-promoting traits” (L95-100).

Line 105: Clarify if the given temperature range is daily average, daily max-min, or seasonal ranges.

We apologize for the lack of clarity. These temperature ranges were daily max and min and we have edited this section to make that clear.

Lines 120-126: Confirm explicitly if the application rates for compost and vermicompost were based on previous research or standard recommendations.

The application rates of 10 MT/ha/yr were based on standard recommendations to achieve an increase of 0.5% SOC over three consecutive years of organic matter application. This information has been added to L137-139.

Lines 133-195: Briefly justify why these particular groups of microorganisms were selected for inclusion in the EM formulation. Were there functional assays conducted prior to selecting these microorganisms?

Prior to the trial, a screen was conducted to assess populations of microorganisms known in the literature to be of agricultural interest for their roles in organic matter decomposition, antagonism against pathogens, and capacity of establishing symbiosis with the root system. Given that all groups were found to be present, a decision was made to include them all to assess their effectiveness as inoculants and persistence in the soil. This information has been added to the Methods section (L144-149).

Lines 279-355: Given that microbial beta diversity effects are minimal, provide a concise justification or hypothesis about why significant agronomic outcomes (pod yields) occurred despite the small effects on microbial community composition overall.

Because this study did not explicitly address the question of the mechanisms involved, we felt it was appropriate to address this hypothesis in the discussion. We note there that impacts on a very few functionally relevant ASVs or groups can lead to improvements in agronomic outcomes even when community composition as a whole is not affected (L606-609).

Lines 433-565: Improve integration of microbial data and agronomic outcomes. Currently, the discussion about specific microbial ASVs (e.g., Trichoderma spp.) and their possible roles in plant growth and productivity could be expanded with stronger references to existing cocoa-specific microbial research.

Thank you for the suggestion. We have expanded the section on Azotobacter and Trichoderma to include more specific information on potential mechanisms of improved growth and productivity based on research in cocoa (L544-559).

Line 446: Discuss briefly whether the observed incremental increase in SOM might have greater significance or cumulative effects in the longer-term management scenarios.

We agree that these results would likely compound over time: “Given the promising trend towards increased SOM even after 15 months in the present study, it is reasonable to assume that applying these amendments over a longer duration would compound these effects, with corresponding benefits for soil health and carbon sequestration” (L511-514).

Line 543: A brief explanation or definition of "cherelle wilt" would enhance reader understanding of pod production constraints.

Cherelle wilt was previously defined briefly as the abortion of immature pods at the cherelle stage. We have added another line to explain the phenomenon: “Cherelle wilt is thus visible when a large number of the immature pods forming after pollination fail to develop further, leaving only the pods that can be supported by available resources” (L616-618).

Reviewer 3 Report

Comments and Suggestions for Authors

The authors show an example of how difficult it is to shift soil microbial equilibria.

The effects they see on soil modification are modest but it's possible that with continual OM /EM amendments a solid positive effect will result.

Several points are raised concerning methods as shown with sticky notes.  A general comment is that the methods require more detail. 

Its interesting that fungal/bacterial ratio  comes up again- a point raised by soil health workers in quite different agroecosystems.

I have concerns about leaf litter  as a control  -  the microbes that colonize their surfaces when on the tree likely would be different from those of composted leaves.  Maybe there were antagonistic chemicals present from leaves  too.  This is found when using different leaf amendments on soils - common with pine needles and cottonwood leaves. 

I notice that lot 1 yield is less than lot 2 ;  but that in lot 1  the less regarded amendment the litter plus EM   brings up the yield to that of lot 2.  Curious   and the low cost here also suggests further studies are needed.  Does the EM  enhance litter breakdown. Litter decomposer preparations are commercial products that claim to enhance plant yields.

Author Response

The authors show an example of how difficult it is to shift soil microbial equilibria.

The effects they see on soil modification are modest but it's possible that with continual OM /EM amendments a solid positive effect will result.

Several points are raised concerning methods as shown with sticky notes.  A general comment is that the methods require more detail. 

Its interesting that fungal/bacterial ratio  comes up again- a point raised by soil health workers in quite different agroecosystems.

I have concerns about leaf litter  as a control  -  the microbes that colonize their surfaces when on the tree likely would be different from those of composted leaves.  Maybe there were antagonistic chemicals present from leaves  too.  This is found when using different leaf amendments on soils - common with pine needles and cottonwood leaves. 

Thank you – we agree with the observations about the difficulty of shifting soil microbial equilibria and the possibility of compounding effects over time, and have made substantial changes to address Reviewer 3’s concerns about methods as outlined in point-by-point responses below. Some details of the microbial methods were proprietary protocols of the external laboratory that could not be disclosed for publication, but the level of detail we have reported is in line with standard practice for papers in this field and the conventions of Microorganisms. Leaf litter served as the control in this study because it is standard practice, whereas incorporation of compost and vermicompost with a motor hoe and the application of a microbial inoculant were novel management practices to be tested. The comparison was thus at the level of management interventions rather than presence or absence of leaves. We have added a line to clarify that leaf litter represents standard agricultural management at this farm (L133-134).

I notice that lot 1 yield is less than lot 2 ;  but that in lot 1  the less regarded amendment the litter plus EM   brings up the yield to that of lot 2.  Curious   and the low cost here also suggests further studies are needed.  Does the EM  enhance litter breakdown. Litter decomposer preparations are commercial products that claim to enhance plant yields.

Lots 1 and 22 are very different in planting design and management, as described in the Methods section, and were included to understand how these treatments might affect yields in different management systems rather than to facilitate comparison of yields between the two lots. We agree wholeheartedly that further studies are needed to investigate this topic further. While explicitly measuring effects on litter breakdown was beyond the scope of this study, we have added a line to the discussion addressing the importance of this mechanism in promoting plant growth and productivity and the need for future research (L514-516).

Specific comments from document:

L54: “can you state though what was seen with mineral fertlizer it could be better or it could have been less stimulatory”

We have rewritten that sentence to make it more clear that the addition of compost to mineral fertilizer had no impact on pod yields, in contrast to the addition of compost to organic fertilizer (L54-55).

L83: “what was the source for all these microbes”

Biofertilizers in the various studies cited in this section were all derived from different sources; that information has been added (L83-84).

L100: “what is the compost made from? does it include cocoa leaves?”

              “Compost was derived from rice chaff, cow manure, and grass” and did not include cocoa leaves (L134-135).

L107: “how patchy is the soil? and what is the character of the soil? is this a healthy or less productive farm? are there known microbial pathogens? are there known toxins such as heavy metals or chemicals from yearly use?”

This information has been added (L111-116): “This area is characterized by regular topography, with minor slopes of up to 3%, leading to relatively homogeneous soils. According to soil maps of Ecuador, soils in the Cerecita Valley are characterized as Inceptisols and vary in texture from clay loam to clay. This is a healthy high-productivity farm without excessive incidence of soil-borne pathogens or heavy metals. As part of the integrated pest management strategy, fosetyl aluminum is applied annually at a rate of 2 kg/ha.”

L118: “could it be the mulch effect v rototillering (air ie oxygen for roots)”

We also considered this hypothesis, which was a primary rationale for measuring soil bulk density after the treatments were applied and incorporated. Given that there were no differences in soil bulk density between treatments, indicating no differences in porosity since density and porosity are inversely related, root access to oxygen would not have been different between treatments either. We have added a line to the discussion to this effect (L499-501).

L123: “did your composts both contain live worms ?”

Neither amendment contained living worms. This information has been added.

L125: “to what depth” [incorporation of compost/vermicompost]

0-15 cm. This information has been added.

L127: “EM?”

Following journal convention, EM was defined at first usage as “efficient microorganisms” (L71-72).

L132: “please explain what this is and reference”

Plating microbial strains and counting colony-forming units is a nearly universal technique in microbial biology, and we have added a widely cited reference for the method.

L134: “of what?”

Of soil solution; this information has been added.

L138: “how many isolates?”

The total number of isolates was not part of the information provided to us by the laboratory.

L140: “need more details what was the pretreatment?”

This is proprietary information that the laboratory was not willing to disclose; we have added a line to clarify that their protocol was proprietary.

L142: “need reference and composition”

The reference with full details of the composition has been added (DSMZ 2007).

L150: “how again details please”

We were unable to obtain details of this proprietary protocol.

L153: “reference”

Reference added.

L155: “how detected mostly its by halos of clearing”

We agree that this is the most common method, but were unable to obtain the proprietary protocol from BioSeb Organics.

L162: “reference what is source of insoluble K?”

Reference added, along with a line specifying that potassium silicate was used.

L167: “all these methods need more detail and references please”

References have been provided for media that were not purchased commercially, and the source has been provided for purchased media. The level of detail provided in this section is consistent with other manuscripts in this journal and the reader is referred to technical protocols for further detail.

L192: “do these have taxonomic and isolate names are cultures lodged into banks as they should be for a publication”

Strains were identified to genus level and remain in the proprietary library of BioSeb Organics, as submitting them to a repository was not included in this standard service contract to isolate the organisms.

L197: “how were the cultures grown and mixed what CFU/ml? were spores present? or just vegetative cells?”

This information was not provided by the laboratory and we have added a line clarifying that the solution was provided by BioSeb Organics.

L202-203: “would this sample include tree roots? what about control soil sample?”

None of the soil samples included tree roots. This information has been added.

L205: “what was the DNA used for?”

For amplicon sequencing; this information has been added.

L210: “of what? soil? microbes?”

Soil; this information has been added.

L306: “of what? how measured?”

Of soil, and measured as reported in the Methods section: “pH was measured in a 1:10 soil:water solution and salinity was measured on a saturated paste using electrical conductance”.

L308: “does not seem to agree with data in table”

Thank you for catching this error. P was significantly higher in the compost treatment; the values were correct but the letters were not.

L308: “is this really significant?”

Yes, the difference in K between OM treatments (treatment means of 1.32 vs. 1.20 vs. 1.15) was statistically significant at p < 0.01, as reported.

L309: “why Cu?”

Copper is commonly included in the analysis of soil macro- and micronutrients and was included in this study because it has been shown to have impacts on soil microbial communities.

L348: “my inference then is that there were native microbes and that these were sensitive to adding more prebiotic materials groupings were the same essentially taking in the scatter”

Yes, neither EM nor OM application had a significant impact on beta diversity of prokaryotic microbial communities, and only OM had a small significant impact on fungal communities, as stated in the legend.

L432: “explain lot in legend”

Added as suggested: “Two lots were sampled in this study: Lot 1, which is 4 ha in size with a traditional full-sun design, and Lot 22, which is 1.5 ha with a mechanized design and includes naturally occurring forest trees.”

L433: “here litter is highest no stat analyses of these yields?”

We have revised Table 3 to show means with standard deviations for annual dry bean yield by treatment and lot, and now present the results of ANOVA in the table caption and results section. There were no statistically significant differences between treatments at the α = 0.05 level.

L446: “did you do an earthworm count on amended soils?”

No, earthworm counts were not conducted at any time in this study.

L460: “again this is pH from soil pore water effects could be greater at the rhizoplane also true for rhizoplane delivery of P and K”

Thank you for this observation. Indeed, rhizoplane effects could be greater relative to effects measured in bulk soil, and we have added this point (L522-523).

L463: “many microbes secrete acids gluconate is an interesting one”

Thank you for pointing this out. We have added to this sentence to make it clear that other microorganisms and other organic acids besides Lactobacillus sp. and lactic acid could be involved (L519-520).

L567: “on what?”

The provisional patent was filed for the microbial inoculant used in this study; this information has been added.

Round 2

Reviewer 3 Report

Comments and Suggestions for Authors

Author Response

Thank you for your interesting comments and suggestions to improve the manuscript. We have revised the manuscript to make the proposed changes in almost all cases, and have explained the rationale for keeping the original wording where we have not. We have also responded to questions and comments that were not tied to suggested revisions here, e.g. to clarify how pod production is regulated in cocoa. Please see below for specific line-by-line responses, noting that line numbers refer to the “track changes” version.

171: reference and what does GYM stand for?

The reference is provided above at first mention (L163). We have added “glucose yeast malt” there as well so the abbreviation is explained at first appearance.

179: hydrolyzed?

Thank you, hydrolysis halo is an accepted term and we have chosen to keep it here.

217-218: and two fungi, T harzianum and T viride

Thank you, revised as suggested.

220: what does the % stand for what is the background solution water or are there nutrients too? any culture products as well?

The microorganisms were prepared in a combined cocktail, which was then diluted in a 2% aqueous solution (i.e. 2% microbial cocktail in water) without additional nutrients or culture products. We have added this information to the Methods section: “This solution was prepared by mixing 4 g each T. viride and T. harzanium (each 3x1010 CFU/g), 1L Lactobacillus sp. (4.3x109 CFU/L), 1L Actinomycetes sp. (1.89x109 CFU/L), 1L phosphorus-solubilizing microorganisms (3x1010 CFU/L), 1L calcium-solubilizing microorganisms (1.26x1011 CFU/L), 1L potassium-solubilizing microorganisms (1.8x1010 CFU/L), 1L Azotobacter sp. (7.5x1010 CFU/L), 1L Azospirillum sp. (1.1x1010 CFU/L), 1L yeasts (1.2x1010 CFU/L), 1L Bacillus sp. (5x1010 CFU/L), 1L Pseudomonas sp. (1.0x1010 CFU/L), and 1L aerobic mesophylls (5x1010 CFU/L), then diluting that combined cocktail 2% (v/v) in water.”

229: spaces please check other places too

Thank you. Revised here and throughout.

239: space please check throughout

Thank you. Revised here and throughout.

251-253: what would you expect the healthiest versus the poorest tree to have for these values need some help with understanding cacao physiology

The flowering index was evaluated on a scale from 1-3, with 1 representing 0-100 floral cushions, 2 representing 100-200 floral cushions, and 3 representing 200 or more floral cushions. The cacao tree is capable of flowering throughout the year, but this generally occurs at relatively low intensity, which would correspond to an index of 1. During peak flowering months (between May and July), a healthy tree could achieve more than 350 floral cushions in a week, corresponding to the maximum value of the index. We have added explanatory notes to this section to clarify that an index of 1 is low to normal flowering intensity and 3 represents peak flowering for a healthy tree.

259: where were these taken from how many replications from one site any pooling of samples from one site?

The sampling and handling methods for these samples are described in section 2.3, and we have added a note to direct the reader to the previous section for more information. There was no pooling of samples for sequencing.

280: based on 16S rDNA sequences and sequences based on using the ITS primer sets

Thank you for the suggestion. We have revised this to read “obtained from [16S/ITS] primers” in both cases.

337: no paragraph change

Revised as suggested. Note that this change is not visible in the track changes version but is changed in the clean version.

340: from amendment with compost than please state what the other two treatments were

Revised as suggested.

345: depends most biology studies you are lucky with 10% reproducibility

Table 2: what is BD?

BD stands for bulk density, and this has been added to the list of abbreviations in the table legend.

Fig 1: good that you could see a change with the increase in trich likely this would also boost bacterial colonizers of the fungi too

Thank you for the observation. We agree.

368: generally paragraphs should be more than one sentence. So collapse into larger blocks for better gramma

We have combined this first sentence with the next paragraph.

Fig 2: good way to show info

Thank you.

414: what was the microbiological load from the vermicompost alone ie before its addition to soil? you would expect it to be different from the compost

We did not sequence the microbial communities in the vermicompost or compost prior to this study, but we agree that we would expect differences between the two communities.

454: sorry for not knowing much about cacao as with other plants flowers may form but these may not go on to form the fruits because of different blockages is that correct?

Yes, that is correct. The flowers begin to develop into small fruits (cherelles), but the tree limits the number of cherelles that develop into fully mature pods based on the amount of photosynthetic assimilates available.

463: are the increases because you have healthier trees do any other parameters change eg chlorophyll content or water withholding in leaves etc?

This is an excellent question, and while we did not measure chlorophyll content or leaf water potential, we note at multiple points in the discussion that more research is needed into the mechanisms involved (e.g. L556, 625).

473: this is a cocoa bean?

Yes, and we have added “cocoa” to clarify.

478: any measure of quality of the bean

No, bean quality parameters were not measured in this study. In cocoa, agronomy studies do not usually assess bean quality because in this industry, agricultural production and postharvest processing are often quite separate.

491: this is the dry beans ? can you clarify?

Compost did not affect pod or bean yields, and we have added “pod or dry bean” to clarify.

522: The soil pH

Revised as suggested.

526: as are the roots exudates which also has protons

We agree.

575: difficult sentence to understand. Please rephrase - split into two?

Revised as suggested.

622: no dash

The dash was part of the tracked changes, showing that a space was added, and does not appear in the manuscript.

624: also more secondary metabs too

We agree.
